# Acceptability of a home-based visual field test (Eyecatcher) for glaucoma home monitoring: a qualitative study of patients' views and experiences

Lee Jones ,[1,2,3] Tamsin Callaghan,[1] Peter Campbell,[1,4] Pete R Jones,[1] Deanna J Taylor ,[1] Daniel S Asfaw,[1] David F Edgar,[1] David P Crabb[1]

[1]Division of Optometry and Visual Sciences, School of Health Sciences, City, University of London, London, UK
[2]Moorfields Eye Hospital NHS Foundation Trust, London, UK
[3]Institute of Ophthalmology, University College London, London, UK
[4]Department of Ophthalmology, Guy's and St Thomas' Hospitals NHS Trust, London, UK

**Correspondence to**
Dr Tamsin Callaghan;
Tamsin.Callaghan@city.ac.uk

## ABSTRACT

**Objectives** To explore the acceptability of home visual field (VF) testing using Eyecatcher among people with glaucoma participating in a 6-month home monitoring pilot study.

**Design** Qualitative study using face-to-face semistructured interviews. Transcripts were analysed using thematic analysis.

**Setting** Participants were recruited in the UK through an advertisement in the International Glaucoma Association (now Glaucoma UK) newsletter.

**Participants** Twenty adults (10 women; median age: 71 years) with a diagnosis of glaucoma were recruited (including open angle and normal tension glaucoma; mean deviation=2.5 to −29.9 dB).

**Results** All participants could successfully perform VF testing at home. Interview data were coded into four overarching themes regarding experiences of undertaking VF home monitoring and attitudes towards its wider implementation in healthcare: (1) comparisons between Eyecatcher and Humphrey Field Analyser (HFA); (2) capability using Eyecatcher; (3) practicalities for effective wider scale implementation; (4) motivations for home monitoring.

**Conclusions** Participants identified a broad range of benefits to VF home monitoring and discussed areas for service improvement. Eyecatcher was compared positively with conventional VF testing using HFA. Home monitoring may be acceptable to at least a subset of people with glaucoma.

## INTRODUCTION

Glaucoma is a chronic progressive condition and a leading cause of non-recoverable vision loss. Worldwide prevalence of glaucoma is projected to reach ~111 million people aged 40–80 years by 2040.[1] Patients with glaucoma require lifelong clinical monitoring; in the UK this is typically within the hospital eye service. High patient caseload has resulted in glaucoma services struggling to accommodate demand; therefore, establishing new methods to support effective surveillance is required. The National Health Service (NHS)

### Strengths and limitations of this study

► This is the first study to collect in-depth qualitative data regarding the acceptability of home visual field monitoring in people with glaucoma.
► Our findings can be used to underpin future research into glaucoma home monitoring and may guide decision-makers when implementing this healthcare service.
► The sample consisted of self-selected volunteers who were members of a glaucoma charity, potentially limiting generalisability of the data.

Long Term Plan, a 10-year strategy describing major practical changes to the NHS service model, aims to make home monitoring of chronic conditions a more widely established and convenient component of patient care.[2] In addition, with the COVID-19 pandemic reducing the number of in-clinic services available, improving remote services and home monitoring is a research priority.[3]

Assessment of functional damage in glaucoma is determined through perimetry, a psychophysical test highlighting areas of reduced visual (luminance contrast) sensitivity. This measurement, known as the visual field (VF) test, is used in combination with other clinical metrics to determine progression of disease, and analysis of serial tests can be used to estimate risk of severe visual loss.[4–6] VF testing may also help predict how glaucoma will impact on patients' quality of life over time.[7]

Home monitoring, whereby disease progression is proactively monitored through patient-led, home-based testing, may help alleviate the increasingly unsustainable burden on hospital eye services.[8 9] Innovation has increased opportunities to perform clinical tests at home.[10] For example, home-based perimetry has been shown to be highly

comparable with conventional hospital-based perimetry[11][12] and may be inexpensive to implement. Other tangible benefits may include increased capacity to collect more frequent VF data, shortened hospital visits and strengthening of the overall glaucoma health service by intelligently prioritising patients.

Previous research suggests people with glaucoma are receptive to forms of home monitoring, such as web-based symptom diaries.[13] However, the acceptability of home-based VF testing has not been explored. This research gap is significant as integration of new healthcare approaches can be challenging and sometimes met with resistance from service users.[14] For home monitoring to be successful, clinical measurements must be accurate, and patients must be accepting of this healthcare model. We assessed accuracy of home VF monitoring in a parallel investigation.[15] The aim of the present study was to investigate the acceptability of VF home monitoring using the Eyecatcher programme. This report focuses on understanding the experiences of patients with glaucoma of VF home monitoring for 6 months, and exploring views regarding wider implementation of this approach in glaucoma care.

# METHOD

## Participants

Participants were recruited via an advertisement in the International Glaucoma Association newsletter (*IGA News*: https://www.glaucoma-association.com/about-the-iga/what-we-do/magazine) on a consecutive basis until the recruitment goal was reached. Participants were assessed for suitability by a glaucoma-accredited optometrist (PC). Twenty people (10 women) aged 62–78 (median: 71) years with an established diagnosis of glaucoma were recruited (table 1). Detail of clinical measurements and ocular findings are published elsewhere (Jones *et al*, 2021). Briefly, all participants recorded baseline best-corrected visual acuity <0.5 logMAR in the better eye. Worse eye (ie, most afflicted) VF loss as determined by Humphrey Field Analyser (HFA) mean deviation (MD) ranging from −2.4 dB (early) to −30.0 dB (advanced), although the majority of eyes exhibited moderate loss (median: −8.9 dB). MD is a summary measure of overall reduction in VF sensitivity relative to a group of healthy age-matched observers, with more negative values indicating more vision loss. Written informed consent was obtained prior to enrolment.

**Table 1** Participant demographics and ocular findings

| Participant ID | Age (years) | Sex | Diagnosis | Ethnicity | Left eye HFA MD | Right eye HFA MD (at baseline) | Received baseline demonstration (Y/N) |
|---|---|---|---|---|---|---|---|
| P01 | 60–69 | Male | PACG | Caucasian | −4.2 | −24.0 | Y |
| P02 | 60–69 | Female | POAG | Caucasian | −2.9 | −9.4 | N |
| P03 | 70–79 | Male | POAG | Caucasian | −2.5 | −3.0 | N |
| P04 | 70–79 | Female | POAG | Caucasian | −3.4 | −7.1 | N |
| P05 | 60–69 | Male | POAG | Caucasian | −21.1 | 0.2 | N |
| P06 | 70–79 | Male | NTG | Caucasian | −8.4 | −10.2 | Y |
| P07 | 70–79 | Female | POAG | Caucasian | −2.6 | −15.2 | Y |
| P08 | 60–69 | Male | NTG | Caucasian | −16.6 | −2.7 | Y |
| P09 | 70–79 | Female | POAG | Caucasian | −14.2 | −1.0 | N |
| P10 | 70–79 | Female | POAG | Caucasian | −30.0 | −13.4 | N |
| P11 | 70–79 | Male | POAG | Caucasian | −22.7 | −11.9 | N |
| P12 | 70–79 | Male | NTG | Caucasian | −3.6 | −3.1 | Y |
| P13 | 60–69 | Female | POAG | Caucasian | −7.9 | −7.1 | N |
| P14 | 60–69 | Male | NTG | Caucasian | −5.4 | −2.4 | Y |
| P15 | 70–79 | Female | NTG | Caucasian | −4.7 | −3.9 | Y |
| P16 | 60–69 | Female | POAG | Caucasian | −4.0 | −0.3 | Y |
| P17 | 60–69 | Male | POAG | Caucasian | −2.7 | −10.0 | Y |
| P18 | 70–79 | Female | NTG | Caucasian | −5.7 | −7.3 | N |
| P19 | 60–69 | Male | SOAG | Caucasian | −3.8 | −0.6 | Y |
| P20 | 70–79 | Female | POAG | Caucasian | −2.4 | −2.0 | N |

HFA MD=average between two monocular tests at baseline.
HFA, Humphrey Field Analyser; MD, mean deviation; NTG, normal tension glaucoma; PACG, primary angle closure glaucoma; POAG, primary open angle glaucoma; SOAG, secondary open angle glaucoma.

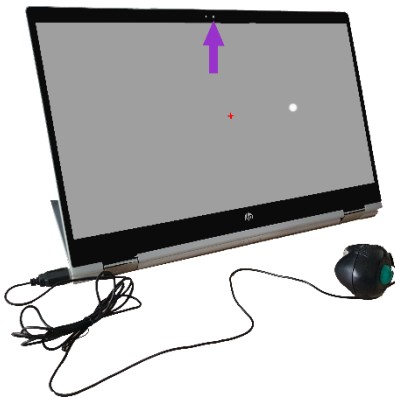

**Figure 1** Eyecatcher testing was performed using home-based perimeter hardware. Participants completed the test monocularly and were asked to respond each time a white dot appeared with a button press using the attached handheld control. The red central point marks the fixation cross. The purple arrow shows the built-in webcam used to observe anomalous tests.

## Procedure

The interviews were conducted as part of a wider investigation of accuracy and adherence to home VF monitoring using Eyecatcher. The full study protocol is described in an earlier report[15] and is briefly summarised here. Participants were asked to perform VF testing using Eyecatcher at home once per eye, per month, for 6 months (12 tests total). Ten participants (50%) were randomly selected to practise Eyecatcher once in each eye, under supervision at baseline (table 1.). The rationale for dividing the sample in this way was to determine whether there were perimetric learning effects using Eyecatcher. We report in our parallel investigation there was no significant difference in mean absolute error between the eyes of participants who received a practice, and those who did not.[15] All participants performed two HFA VF tests in each eye (24-2 SITA Fast) at baseline and again after the 6-month home VF monitoring period was completed at City, University of London. All participants received email reminders when the tests were due and were provided with contact information for any queries. Eyecatcher VF tests were performed using a custom screen perimeter on a convertible laptop/tablet (figure 1.). The test was a variant of the 'Eyecatcher' test described previously[12 16] and closely mimicked conventional static threshold perimetry. Good concordance between VF tests at home using Eyecatcher

and in the clinic using HFA perimetry was observed (figure 2). To date, the development of Eyecatcher has been funded in part by university research time and non-commercial research grants. The complete source code is available online (https://github.com/petejonze/eyecatcher) under a non-commercial (General Public License V.3.0) licence.

## Data collection

Semistructured interviews were conducted between January and March 2020 on completion of the 6-month home monitoring period. All but one of the interviews were carried out at City, University of London. Due to transport restrictions during the COVID-19 pandemic, one interview was performed via telephone. Interviews were carried out one-to-one by a male research fellow with a background in psychology (LJ; n=19) or a male glaucoma specialist optometrist (PC; n=1). Interview duration ranged between 6 and 38 (median: 17) min. The interview topic guide (table 2) included questions on participants' general attitudes toward home VF testing and questions regarding specific behaviours such as how VF home monitoring could be carried out and why. To understand key determinants of behaviour, the questions were guided by the COM-B model (Capability, Opportunity, and Motivation for Behaviour) of the Behaviour Change Wheel framework.[17] The framework was used to determine how VF home monitoring might be affected by individuals' knowledge, skill, opportunity and motivations.

## Data analysis

Audio-recordings were transcribed verbatim by a professional transcription service. LJ performed thematic analysis as described by Braun and Clarke,[18] whereby transcribed texts were read and re-read to ensure familiarity, and meaningful units were coded using primarily an inductive approach (ie, driven by the content of the data). A deductive approach was later used when considering the identified themes in relation to the COM-B framework which was used to collect subthemes into the respective key themes. Transcripts were coded at a semantic level, considering the explicit content of the data. Data were analysed using NVivo V.12 (QSR International, Cambridge, Massachusetts, USA). Data saturation was determined post hoc and indicated further data collection or analysis was unnecessary. The coding framework was independently reviewed for suitability by a research optometrist (TC),

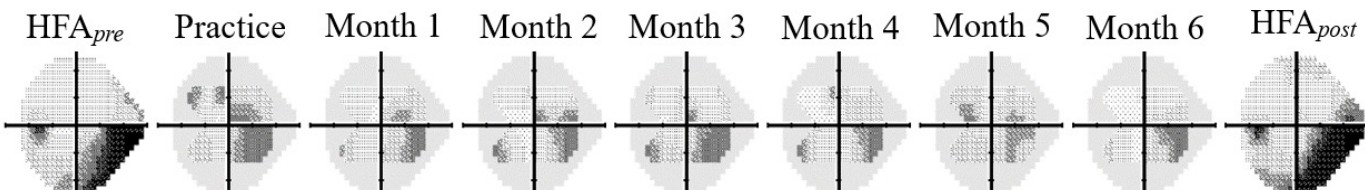

**Figure 2** Visual field (VF) testing regimen. Data shown are raw VF results at each test interval for one randomly selected eye. Darker tones represent decreasing visual sensitivity. The first and last columns show mean averaged data from two 'pre' and two 'post' reference tests, performed in clinic using a Humphrey Field Analyser (HFA). As described elsewhere,[13] there was good concordance between Eyecatcher and HFA.

**Table 2** Interview topic guide

| COM-B model | Theme | Question |
|---|---|---|
| Capability | Psychological | How confident were you using the device at home? Was it easy? Was it difficult? Did you become more confident using it over time? |
| | Physical | How do you feel you performed on the home visual field test compared with the test in a clinic? Do you think you performed differently between the two? |
| Opportunity | Social | When performing the visual field test at home, were you usually alone or were other people around? How did you find this compared with performing the test in a clinic? |
| | Physical | How would you describe performing the visual field test in your home setting? Were there any challenges caused by performing the test at home, specifically? Were there any benefits about doing the test at home, as opposed to in a clinic? |
| Motivation | Automatic | What, if anything, would encourage you to consistently complete the visual field test at home? What, if anything, would prevent you from using the visual field test at home consistently? |
| | Reflective | What do you think is the value of performing the visual field test at home? For whom do you think it is most appropriate to use home visual field testing? If you were to do the visual field test at home again, what would you want to do differently? What would you like to keep the same? |

COM-B model, Capability, Opportunity, and Motivation for Behaviour.

who interpreted patterns in the identified themes and suggested amendments. Where there were differences in coding choices (34 occasions in 558 coded references), there was discussion with assistance of a third reviewer (DJT), and when consensus was reached the coding framework was finalised. At this point, a member-checking exercise was employed to assess the credibility and authenticity of the final coding framework. In line with recommended practice,[19] all participants were emailed a word document of the coding framework which included the main themes and subthemes and anonymised quotations as illustrated in the results. Participants were given the opportunity to clarify or elaborate on any aspect of the framework and to describe if the interpretation was congruent with their experience. The process allowed for an assessment of descriptive validity (factual accuracy of accounts) and interpretive validity (accuracy in determining meaning of quotes). Of 20 participants, 14 responded during this exercise and the manuscript was amended following recommendations. Finally, the study was designed and reported following the guidance of the Consolidated Criteria for Reporting Qualitative Research.[20]

## Patient and public involvement

The purpose and design of the study were described in a newsletter article from a charity for people with glaucoma (IGA), and the study outcomes will be disseminated to the wider glaucoma community in the same way. Three patients with glaucoma known to the research team reviewed the study design during the planning phase and provided feedback during a focus group. Study participants helped to steer this project during data analysis and reporting through the member-check exercise. Participant feedback will also help improve the design of future home VF monitoring studies.

## RESULTS

All 20 participants enrolled in the study took part in a semistructured interview. Nineteen of 20 participants completed the full regimen of monthly monocular VF assessments in each eye for 6 months. One participant (ID P20; see table 1) discontinued after four sessions. Interview data were coded into four overarching themes relating to experiences of undertaking VF monitoring at home and attitudes towards its wider implementation in healthcare: (1) comparisons between Eyecatcher and HFA; (2) capability using Eyecatcher; (3) practicalities for effective wider scale implementation; (4) motivations for home monitoring. Key themes and subthemes are summarised in figure 3.

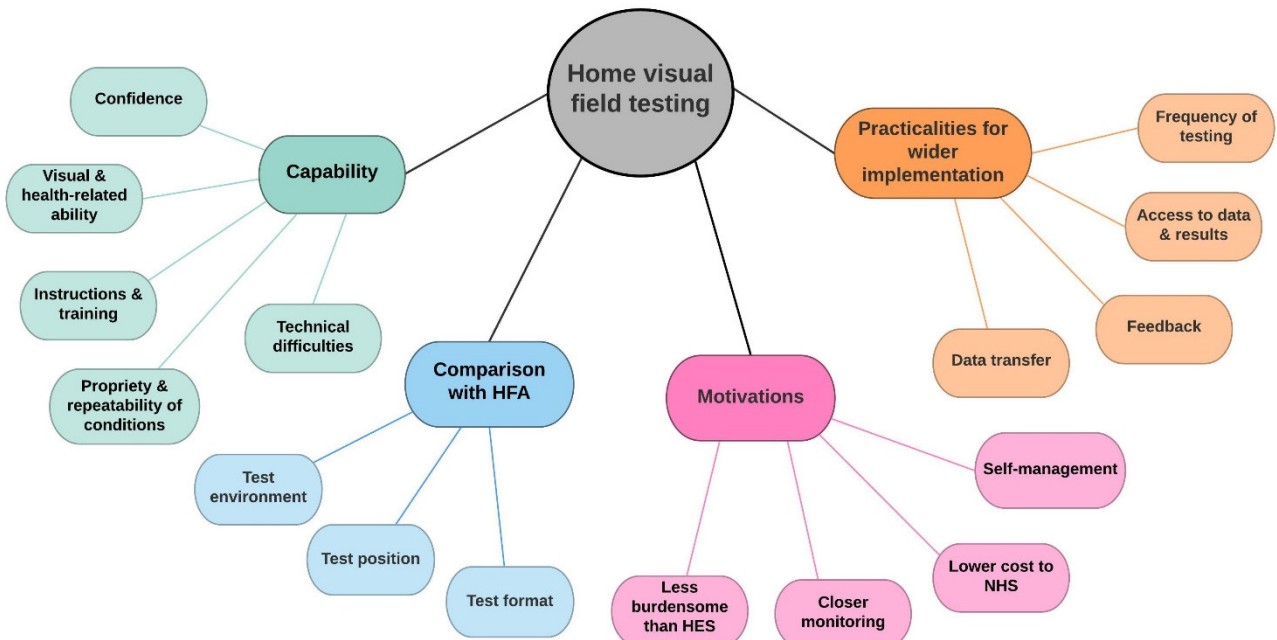

**Figure 3** Coding tree showing main themes and subthemes generated from the analysis, and how different categories relate to each other. HES, hospital eye services; HFA, Humphrey Field Analyser; NHS, National Health Service.

### Theme 1: comparisons between Eyecatcher and HFA
#### Testing position
Participants reported one of the biggest differences between Eyecatcher and conventional perimetry was the body position required for testing. Eyecatcher was seen to permit a more natural body position due to the removal of the chin rest and by allowing free head movement:

> I find with the optician tests [HFA], I find it very difficult, it never seems possible to get my chin and my body and my head in the right place on it. So I'm always left feeling very uncomfortable during the test. So not having that to contend with was quite good. (P03)

Although the freedom of Eyecatcher testing was appreciated, questions arose whether an unfixed testing position could impact the accuracy of measurements:

> While I was taking the test, I was thinking have I moved? Am I still in the right place? (P07)

A dislike of the enclosed environment associated with conventional perimetry (ie, perimeter 'bowl') was observed, and preferences for Eyecatcher's unrestricted testing method were reported:

> Well I don't actually like putting my head in that round ball thing it's claustrophobic. Because if they [HFA and Eyecatcher] are very much alike then it would be a much better option in my view to be home based doing it. (P08)

However, removal of perimeter bowl raised queries about distraction from peripheral stimuli present in the field of vision:

> Looking at a laptop, there are other things within the field of vision in one's own home. For example, the settee or the television. When I was doing it, I was thinking I can see the settee. Also, in an eye where you can't see very well anyway, it felt like added distraction. (P02)

#### Testing environment
Performing VF tests outside a busy clinical environment had perceived advantages. For example, hospital testing environments were considered noisy and distracting, whereas testing at home allowed for better control of these factors:

> One of the things with doing it in the hospital is that sometimes you'll get a neighbouring patient who doesn't really know what they're doing. That's unfortunate, and that can be quite distracting, when you've got a lot of chat going on in the background, and you're trying to concentrate, so obviously, doing it at home is far, far better. (P13)

Eyecatcher visual fields were viewed as more relaxing due to the home environment, and it was suggested such measurements would likely better reflect visual function:

> I think it's much more comfortable, much more relaxing, much easier. (P08)

> It's [Eyecatcher] certainly a better experience. I felt as if the results should reflect my visual field better. I've always been a bit sceptical about the old - well, the current machines. (P18)

The flexibility around scheduling tests at a convenient time was welcomed:

> I can do it when I want to do it. If I've got a spare half hour I can fit it in around other things that you're doing in your life. So from that point of view you haven't got an appointment where you've got to be on a certain day at a certain time so I prefer that. I think it's because you call the shots, you decide when you do it, where you do it, and how you do it. So as long as the technicalities are clear it's much more flexible which I like, yes. (P08)

### Test format

Unlike conventional perimetry, Eyecatcher uses visual feedback (a confirmation flashing dot) indicating the true stimulus location after each button press. Participants had mixed opinions regarding this feedback. Visual feedback to confirm the stimulus was correctly identified was sometimes welcomed:

> The great thing about this [Eyecatcher] was having the little white discs appear when you press the button - it makes a huge difference. With these little white discs appearing, I know that I've seen the white dot and I've correctly identified… So that does help enormously. (P01)

In other cases, the utility of this feedback was questioned, and participants described how it became distracting when the stimulus appeared in an unexpected location:

> The little feedback flash - Now occasionally I would get feedback not in the place where I thought I could see the spot so that was a bit distracting, I think. It interrupted your flow. (P05)

> Sometimes I'd click thinking I'd seen a light in one place and the confirmation light came in a different place, I found that a bit disconcerting. (P15)

Participants had mixed interpretations regarding the length of Eyecatcher tests. One participant regarded Eyecatcher as the faster test:

> I thought I probably did better than I do in the clinic environment, which may be because I could concentrate better and I think the test was a little bit shorter as well. (P11)

In two cases, Eyecatcher tests were perceived to be longer or more variable compared with conventional perimetry:

> The tests were of quite variable length, so I didn't know whether they were supposed to be that way or because I'd pressed something I shouldn't have pressed. (P09)

> They seem to go on for a very long time. In the end, certainly by the last one, I said, you've pressed enough. That's it. You've already done - I've counted

300 attempts, you know, just leave it, you're getting too tired to do it. (P12)

## Theme 2: capability using Eyecatcher
### Confidence

There was generally a high level of confidence in using Eyecatcher, and this was attributed to familiarity with using laptops and other computer devices:

> Well I was fairly confident doing that [Eyecatcher]. I took computer programming years ago, so I've grown up with computers, so that was fine. (P07)

Even among those who self-described as having limited computer literacy, confidence performing the test remained:

> I was confident at performing the actual physical side of the test. I'm fairly limited with my IT ability and it was that which really caused me to be a little apprehensive. I had no problem with setting up the test and doing it. (P06)

Confidence using Eyecatcher was also attributed to prior experience and familiarity with VF testing on the whole:

> I just thought it was fairly straightforward to do. I've done quite a few field tests in my time, so it was just like normal, really, for me. (P04)

For some participants, confidence in the Eyecatcher testing procedure accumulated after the initial tests:

> It maybe took a couple just to get used to it, but it was fine after that. It's quite simple. (P18)

### Instruction and training

Clear testing instructions were an important resource to remind participants how to perform Eyecatcher tests:

> The instructions were quite clear and it was quite easy to set up a laptop and start it off. I referred back to the laminated instructions each time, which was helpful, because I had forgotten one or two parts of it. (P01)

Half of the sample received a practice trial using Eyecatcher at baseline, which appeared to be highly valued. These participants described how an increased period of training at the start of the trial would be beneficial in improving familiarity and generating confidence:

> Familiarity helped, once I got the hang of it. I thought it would have been perhaps better to have more of a run-through before we got started, just to get the hang of it; feel that bit more confident a bit earlier in the experience. (P12)

In some cases, insufficient training in using Eyecatcher resulted in a lack of confidence and apprehension around the test, and this was considered a barrier to

home monitoring, particularly for those who may be less familiar with technology:

> People don't realise what a difficulty it is for older people like myself, who become apprehensive. The simplest aspect of IT you become apprehensive and worried about. It has to be step by step, it's no good assuming that I know every single step. It has to be set out step by step, otherwise I fail. (P06)

> You need to be able to operate, albeit in a fairly limited way, a PC and be aware of that basic bit of equipment which I think some very old people might struggle with possibly. (P17)

### Propriety and repeatability of testing conditions

Overall, the biggest challenge for participants was ensuring the home environment was appropriate for conducting Eyecatcher tests. While all participants could execute the test accurately, some raised concerns whether environmental variations would impact measurements:

> The bit that still worries me is the bit about getting the lighting right in the room, because there were no specific instructions as to how to do that. Certainly I found that it varied from day to day and time to time. I don't know at what level that impacts the results of the test. (P03)

> I can't tell you that they were identical at all because the lighting I found difficult. I had to move the laptop around a bit and then when the darker nights arrived, because I always did it in the evening, that was easier but then I found I couldn't see enough. (P20)

As well as difficulties optimising luminance, a further challenge was ensuring the laptop was at the correct height and distance for testing:

> I was more paranoid about getting it to the right height, and sitting in the right place and being the right distance away from it, and those kinds of things. (P07)

> Setting it up the first time is really quite tricky because if you do it at a normal work surface height, it's too low. You've got to raise the machinery, so to speak or the – something to put the laptop on. So just fiddling around with that and trying to get yourself the right distance back and your chair in place, that's quite tricky to do. (P10)

However, it was also acknowledged that testing conditions for conventional perimetry are often not ideal, for example, the auditory disturbances described earlier. Creative methods were described for attempting to best ensure repeatability of test conditions:

> Once I got the setup done and photographed it, I then thought well I'll be able to do this next time, I'll just do the same setup. It meant moving things around at home and then putting them back again.

> I photographed it and thought, well I can replicate this scene. (P02)

Although it was possible for all participants to perform Eyecatcher under suitable conditions, more flexible testing conditions or alternative hardware may be needed for wider scale implementation, such as the use of a head-mounted display:

> The equipment itself has to change, surely. Piling up a laptop on wobbly books is not a way to go forward. You need a piece of kit that you can accurately get your distance that is not left to a human to set it up. (P16)

> The ideal thing would be something you can actually put on your eyes, it blanks out one side and the whole thing works. Then you would take away some of those problem with light and everything else. (P03)

### Visual and health-related ability

While participants described relative ease in completing the test, there was increased difficulty initiating the test for those with reduced vision in at least one eye. Problems arose when setting up the test while also having one eye occluded:

> I started the test with the right eye first, I've lost vision in the middle of the right eye, so it's difficult to read the instructions on the screen, so that was a bit awkward to start the test going. (P01)

One participant described how using Eyecatcher led to increased symptoms of underlying vertigo. As a result, this participant discontinued testing after 4 months:

> It didn't really suit me. I have vertigo anyway from time to time, and I found that after every session of doing the visual field tests I was terribly dizzy. The usual, felt a bit sick and couldn't get my balance properly. (P20)

### Technical difficulties

A technical issue occurred whereby some devices required a password update. This fault was due to a software update which would not occur outside the context of this exploratory study. While this was straightforward to resolve, participants acknowledged this may be difficult for some to rectify:

> I had a problem with the password. Because I'm fairly familiar with computers and PCs and passwords that wasn't a real problem, but I could understand that some people might become worried they'd broken it or something. (P17)

One solution was to develop a bespoke testing system where the sole purpose of the hardware is to perform the VF test:

> If it had been programmed just to do one thing. But for instance, once I got into completely the wrong program, unbeknown. I don't what I did. (P10)

A helpline to accommodate for software queries and other technical difficulties was available during office hours to all participants. One participant discussed its importance:

> I did ensure that when I did the test it was on a workday, like a Monday to Friday because I thought there was a better chance of someone being at the end of a phone to help me get out of a pickle than there would be if I did it on a Saturday or a Sunday. I think I always would want to know that there was a helpline that I could ring up in case something went wrong with the laptop. (P02)

### Theme 3: practicalities for effective wider scale implementation
#### Feedback
Two forms of important feedback on the test were described: verbal feedback on test performance as well as feedback regarding confirmation of successful completion of each test. Verbal feedback from the perimetrist when undertaking conventional perimetry provides valuable encouragement that the test is being performed correctly, and participants commented on the impact of absence of immediate feedback:

> One of the things I found problematic was there didn't seem to be feedback. I felt a bit unfulfilled from that point of view. One of the differences between having it done at an optician's and doing it at home is that at the optician's, you get immediate feedback. Some sort of dialogue, to talk about it, to unpack it a bit more. (P12)

It was suggested that test-by-test feedback to confirm Eyecatcher tests had been appropriately performed and data had been received and stored successfully would be a valuable addition to home monitoring:

> Having done that [Eyecatcher], you should then get an email on completion to say thanks for doing it. Just make it a bit two-way. I think that would be really good. I think that would encourage people to engage with it more. (P14)

Comparisons were made to other health-related clinic visits, such as blood donations. Greater transparency regarding the location of the test information was desirable:

> When I've donated [blood] I get an email to say thank you. Then a few days later I get an email to say where my blood went to. It's quite nice to get the two-way traffic. (P14)

#### Access to data and test results
The current home testing system did not allow participants to retrieve their VF measurement data. Ability to access Eyecatcher outputs was considered of high importance to some participants:

From the point of view of the patient, if I wasn't to get the results until six months later, that's when I get them anyway so there's no particular point. (P09)

> It would be really nice if I could have printed the results out. But that's just me. I can understand that some people wouldn't want to do that. (P14)

However, some participants noted the possible psychological effects of accessing the output:

> If one was being asked to do it [Eyecatcher] as part of your treatment plan, it would be quite good to see the result. I suppose the only difficulty is that it panics some people. If you look at something without really understanding what you're looking at. They are a bit peculiar and that might freak you out a bit I suppose, which maybe isn't all terribly helpful when you're a layperson. (P13)

#### Data transfer
In the current study, participants returned the laptop device at the end of the trial where data were extracted. Questions were raised over practicalities of transferring Eyecatcher data to clinical review.

> How is the data on the device going to be communicated? Will that be done in real time and how will that information get to the hospital? Is that going to be done by electronic transfer or emailing it? (P02)

> Presumably, if this [Eyecatcher] came into common use, it would be linked in some way to your medical record, I assume the results would work with the hospital record. (P13)

A possible barrier to electronic transfer of home VF measurements was the need for an internet connection, and how individuals may not have access to this service:

> I've very intermittent wi-fi connection at home and so that adds another layer of complexity. If it's ever going to be an interactive process where you do need a wi-fi connection, I can see problems. (P02)

#### Frequency of testing
Eyecatcher testing was performed once per month for this exploratory study. Some participants questioned the need for such regular testing intervals:

> The challenge for me - because it had to be done every month, I found was too repetitive because normally you don't have your visual fields done every month. (P16)

For other participants, increased regularity in performing the test was viewed positively, and this was perceived as more closely monitoring visual function. This is discussed further in theme 4.

### Theme 4: motivations for home monitoring
#### Closer monitoring of eye health
While questions were raised about the necessity of monthly testing, home monitoring was viewed as a potential to

complete more regular VF tests, enabling closer monitoring of visual function:

> I appreciate how important they [visual fields] are and how important they are to get a good result out of them. I think it is easy to broaden that and collect as much data as you can. Because I know that's a really good early indicator of things deteriorating. I think it is very useful thing to do. I'm fully with it. (P14)

> I know that glaucoma gets worse over time and I know that it's important to do everything to monitor the progression and if something were deteriorating, I'd like it picked up sooner rather than later. I can see that doing it [visual fields] frequently, aids that process. (P02)

It was acknowledged that more frequent testing may help to mitigate issues regarding measurement accuracy. For example, there was a narrative of 'getting the test right' as often there may be a long time interval before the test will be repeated:

> If there was a sudden change, I could get onto a consultant quickly rather than waiting six months. Or at least that I would know it and at the next month's see if it was really a sudden change or if I've just been too sleepy the last time I did it. (P09)

On a number of occasions, participants spoke openly about their fears the VF test is not performed as regularly as they would like in the hospital eye service:

> I don't seem to be getting regular appointments. I need to keep phoning up to get my appointments at the hospital, I have six monthly checks and it's been a year coming up in March since I have been. I like to have regular field tests to make sure I'm not losing any sight, and I think that's quite worrying, if you've got this [Eyecatcher] at home, at least if that's being monitored by people, it's not such a worry. (P04)

Although closer monitoring through more frequent testing was welcomed, the point was raised that VF tests are just one component of a comprehensive glaucoma examination:

> If I go to the glaucoma clinic, there is usually more than this test. So, in actual fact, if I'm just isolating to this type of test and not going to the hospital, then I'm losing out on other tests, so it might be one test of several. (P06)

### Lower cost to NHS

It was important to participants that home monitoring may be a viable means of cost-saving to the NHS. This was due to the belief Eyecatcher was time-saving and less labour-intensive compared with conventional perimetry:

> There's the cost of the NHS staff to facilitate the test, I imagine there'd be a big saving on money for the NHS, which is always good. (P19)

> It [Eyecatcher] has to be the way forward rather than sitting for hours in hospital clinics, which is wasting your time, but also the time of all the staff. From my point of view, I think that's a huge benefit. If you think about the way we afford an NHS, we can't afford it unless we do things like this. (P10)

### Less burdensome than appointment at hospital eye service

Traditional glaucoma-related hospital visits were considered time-consuming, and participants suggested home VF testing could partly alleviate the burden associated with outpatient activity:

> Presumably I still have to come into a clinic at some stage, but it might reduce the requirement for that. It is quite an effort to go to an appointment at the hospital, it does seem to take a lot of time really. (P03)

Home monitoring was viewed as particularly promising when considering the benefit to individuals with less accessibility to healthcare due to restricted mobility:

> I can see for people who have limited mobility or if they have a disability of some kind it would be a great bonus to be able to do it from home. (P08)

> Anyone who's got mobility issues, it would be ideal if they were able to do it at home. (P17)

Other individuals most likely to observe benefit were those currently requiring two hospital attendances due to unsynchronised clinic appointments and consultations. The ability to complete VF tests at home prior to consultation was viewed as a pragmatic approach to glaucoma monitoring:

> I get two separate appointments - one for the fields and one for the review. I don't have a choice of the dates, they just send me a letter and that's it. Sometimes I've gone in to review and I haven't had the fields test done so it's been a waste of a review. So I think doing it at home is really good because you can do it when it's convenient for you and also you can make sure you've got a fields test before you go in for a review. (P05)

> They give me two appointments, one is for the visual field test and maybe another test as well and then I come in again to see the doctor. So I suppose doing it [Eyecatcher] at home would cut it down to one visit instead of two. (P01)

Enthusiasm for home monitoring was also amplified by the prospect of reduced need for travel and the associated benefit to the environment:

> Of course you don't have to go anywhere. You're doing it at home so good for the planet, less travel. (P17)

### Self-management

Opportunity for home monitoring afforded some participants a sense of control of their glaucoma care which was highly valued. In comparison with hospital attendances,

home monitoring was described as having potential to create a more sustainable way of living with glaucoma:

> I had a sense of control, and I think you're not put in the position that you are in a hospital environment where patients are sat in rows and treated, inevitably, like some sort of factory unit. So you feel that it's the flip side of the responsibility, which is frightening, but it also encourages you to take responsibility for yourself in the same way as using your drops. That's good, I think that you feel that there's something you can do. I get a glow out of thinking that I might make it easier for somebody else in the future. (P10)

However, it was apparent that not everyone would view self-management in a positive manner. For example, home testing requires a high degree of patient activation:

> I'm not sure everyone would be suitable for doing a home test. I think you have to be fairly disciplined, and probably quite experienced in doing them, so that you know really what's supposed to happen. (P13)

In some cases, the idea of self-management drew negative connotations. Specifically, transference of responsibility to the patient was sometimes undesirable and led to questions about individuals' own role in their glaucoma care:

> It's much easier for me to walk into an optician's or a hospital, have everything laid out, just sit down and do it. So, the onus isn't on the patient. (P16)

> I found the responsibility for getting the setup right is shifted from the people in the hospital who know what they're doing to a novice like me who was just hoping I was getting it right. (P02)

## DISCUSSION

This is the first study to investigate service users' experiences of participating in a VF home monitoring programme. Self-selecting members of a glaucoma patient organisation recognised the usefulness of testing and valued the opportunity to have their condition monitored from home. The findings suggest some individuals with glaucoma may embrace wider implementation of VF home monitoring. The significance of these findings is highlighted with respect to the COVID-19 pandemic, where conventional hospital-related outpatient activity cannot be easily achieved, and the potential gains of glaucoma home monitoring are further emphasised.[21]

In general, participants welcomed the opportunity to complete more regular VF tests, as increased measurement frequency was viewed as closer monitoring of glaucoma-related health. There was confidence that deterioration in visual function could be identified more rapidly, allowing changes in treatment if required. This resonated particularly among those with glaucoma under annual review, where there was candid discussion about

concerns over delays in receiving appointments and lengthy time intervals between measurements. Delayed follow-up due to hospital-initiated appointment rescheduling has been reported as a significant cause of VF loss in patients with glaucoma,[22] and use of VF home testing may help to mitigate such instances in future. However, some participants questioned the necessity of monthly VF measurements, and although acceptable in the context of a research study, they risk becoming a nuisance if required long term. Our rationale for monthly testing was linked to investigating feasibility, as opposed to determining clinical need. Multiple studies have shown that intensive VF testing can help identify and prioritise individuals most at risk of rapid sight loss,[4–6] and there is convincing evidence that three VF tests per year provide adequate information to aid clinical decision-making.[23] Further assessment of the optimal number of home VF measurements is required. Applying a targeted approach to patient selection, where home monitoring of those at greatest risk of debilitating visual loss based on individual's age,[4] condition,[24–26] ethnicity[1] or socioeconomic status,[27] could maximise the potential benefits of home VF monitoring.

A further concern regarding the conventional approach to VF testing was the issue of performance pressure, and how limited opportunities to perform the test gave credence to beliefs about needing to 'get the test right'. Specifically, concerns about how lethargy might impact on measurement outcomes were reported, and similar findings have been observed elsewhere.[28] Greater capacity to highlight spurious results through more frequent testing was considered an advantage. Similarly, home testing allowed participants to complete measurements at their convenience. It was thought that testing in these circumstances would likely better reflect true visual function, as extraneous variables such as energy levels could be better controlled. For example, participants could complete tests when feeling at peak performance using Eyecatcher, whereas conventional testing relies on designated testing times. Indeed, transient factors such as fatigue, lack of concentration, or delayed reaction time can significantly obscure VF measurements and increase variability.[29] Although our findings suggest participants exhibited similar testing behaviour and outcomes between Eyecatcher and HFA,[15] we highlight that flexibility around performing VF tests is of value to people with glaucoma, representing a more quality-oriented approach toward individual patient needs and preferences.

Participants generally compared Eyecatcher positively with conventional perimetry. In particular, the absence of chin and head restraints with Eyecatcher was particularly preferred over the HFA, which was viewed as an uncomfortable and unnatural testing procedure. There was a diversity of opinions, however, and some participants did question whether Eyecatcher's more relaxed testing procedure, and/or the lack of a conventional perimetric bowl (to occlude potential distractions), may result in more distractions and less reliable measurements. Our

quantitative analysis suggests the home monitoring data were generally of good quality, and that any loss of reliability was largely compensated for by the overall increase in the volume of data.[15] However, distractions will always be a concern for any unsupervised/uncontrolled testing. In response to this, we have developed computer-vision, AI technology to autonomously detect and account for lapses in concentration. This technology uses data from the web cam on the portable device and can be embedded within Eyecatcher. Our investigation of this technology in a large group of people, although with healthy vision, is the subject of a report published elsewhere.[30] More generally, the fact that there were mixed opinions regarding the Eyecatcher interface (ie, with some finding the removal of the bowl liberating, while others expressed concerns regarding increased distractions) highlights the inherent complexities associated with developing and implementing novel health-related technologies and ensuring they match the needs and preferences of the target users. We hope that the in-depth feedback generated in this study will serve as a useful reference for clinicians, researchers and commissioners working in the area of VF home monitoring.

Several participants noted that test durations were often longer and more variable than with standard automated perimetry. This is true. Median test durations were 4.5 min vs 3.9 min for HFA (SITA Fast), and a minority of individual tests lasted as long as 10 min.[15] These differences are not intrinsic to home monitoring, but rather reflect the fact that Eyecatcher is at present a crude prototype, that has not been optimised in the same way as modern commercial perimeters (see Jones et al for detailed technical suggestions[15]). It is highly encouraging though that people nevertheless found home monitoring positive and desirable, despite current technical shortcomings.

Participants recognised the possible economic value of home monitoring, especially in terms of saving costs in the NHS. In particular, time and resource-saving potential of remote testing was viewed very positively. Previous research suggests people with glaucoma generally approve of service-level cost-saving, so long as it is in tandem with improved efficiency and efficacy.[31] Moreover, perceived financial worth of home testing may serve as behavioural encouragement to adhere to prescribed testing regimens. Conversely, it should be noted that home monitoring itself has associated costs, and the extent to which home monitoring of VFs is cost-effective is yet to be established.[32] Prospective costs associated with VF home monitoring include data management and the cost of supplying and maintaining any associated hardware (though exactly what hardware is required, or whether a perimetry test could even be provided simply as a 'downloadable app' remains an unresolved question and highly contentious). Home monitoring provides opportunity for increased frequency of VF testing, which may place a burden on clinicians. Considering previous estimations of costs associated with conventional VF testing,[33] we predict home VF testing will prove cost-effective, at least for the most

high-risk patients (eg, younger individuals or those with risk factors for fast progression). A formal economic evaluation is required, however.

In addition to service-level economic value, participants associated home VF testing with personal savings such as reduced transportation to clinic which can be costly.[34] Financial aspects of home monitoring are particularly relevant to those with unsynchronised testing and consultation appointments, which usually duplicates costs. This finding emphasises the potential gains associated with home monitoring in ophthalmology, including direct financial savings to patients. Indirect costs to patients associated with hospital visits included the general burden of lengthy hospital visits which typically rely on assistance from an informal caregiver. Evidence suggests companionship during eye-related hospital outpatient visits may have substantial 'unseen' consequences, such as imposed strain on social relationships.[35 36] While home monitoring will not obviate the need for hospital attendances, participants viewed it as an opportunity to streamline the glaucoma service. Home monitoring can be useful to augment conventional outpatient activity and help indicate when other glaucoma-related measurements need to be prioritised. It is noteworthy that when comparing home monitoring of their VFs with monitoring in a clinic environment, participants sometimes referred to VF tests conducted in/by opticians in addition to those carried out in the hospital eye service. This is unsurprising given that when people with glaucoma attend their community optometrist for a routine sight test, a VF test (often using the same or similar equipment to that used in hospital eye service) is normally carried out.

Eighteen (90%) participants discussed their experiences of ensuring the VF tests were performed under the appropriate conditions at home, such as under dim lighting while positioned at eye level with the laptop perimeter. Despite all participants successfully completing VF tests at home and to a good standard, there were some concerns about the propriety and repeatability of the testing conditions. Our parallel investigation demonstrated no relationship between changes in illuminance and changes in VF test score, or between absolute illuminance and absolute test score, suggesting luminance had little effect on the study data.[15] It is possible that a period of training to familiarise with Eyecatcher may reassure patients about the testing conditions and procedure. However, there appeared to be no major differences in difficulties experienced throughout the study between those receiving a demonstration at baseline compared with those without.

Wider scale implementation of home VF testing attracted questions about transference of test data to care providers and storage within patient clinical records. In the current study, remote access to the laptop perimeters was disabled, thus data extraction was only possible upon returning the device at the end of the study. A system automatically expediting data for clinical review could provide a more pragmatic approach to VF home monitoring, and has been adopted in

remote monitoring of other health conditions.[37] Previous approaches to portable perimetry have developed applications (eg, Melbourne Rapid Fields[11]) notifying treating clinicians if a significant change in vision has been detected. Inclusion of a remote data access feature would be relatively straightforward to implement in future studies. Our participants acknowledged barriers relating to transference of data, stating this should be an automated process and not require additional responsibilities for the service user. It would be pragmatic to use a device compatible with internet accessibility whereby data are transferred direct for clinical review.

Our sample contained a range of older adult ages (62–78 years), and the vast majority reported ease using Eyecatcher. It may be argued that individuals in the later stages of life would not necessarily benefit from increased monitoring, because the likelihood of patients suffering with visual impairment is linked to the rate of glaucomatous progression and extent of VF loss at presentation.[4] There was a belief among participants that some elderly people with glaucoma may struggle with using home monitoring technology. Evidence suggests older people engage better with technologies that are considered to have value, including those enabling self-monitoring.[38] Patterns of technology usage suggest older ophthalmic patients frequently use technology, for example, 88.5% of people aged 66–79 years attending Manchester Royal Eye Hospital report regularly using technologies including desktop and laptop computers.[39] Many older adults aged 65 years and over use devices relating to healthcare, such as glucose and cardiac monitoring tools.[40] It is perhaps most appropriate to expect large variability in abilities relating to technology among older populations. Indeed, increased variation in physical and cognitive function as people age requires healthcare policies to consider service users on a more individual basis.[41] Thus, a patient self-select approach may be the most practical means of best ensuring effective implementation of home VF monitoring. As one participant in this study described it, 'I would be very loath to rule people out, and people surprise you all the time' (P10).

Issues relating to inability to view measurement results were raised by one participant. As the name would suggest, home monitoring devices should allow users the opportunity to view measurement outcomes. In this study, participants did not have access to their study data, leading to reduced motivation to complete the test for this participant. Access to test results is important to patients,[42] and people with glaucoma are often very interested to learn about their VF results.[28] This raises a number of important questions around what information would be meaningful, useful and actionable for patients. Evidence suggests direct release of test results is associated with increased healthcare engagement.[43] However, unintended consequences which must be considered include increased anxiety regarding unusual results,[13] and the ethical implications of providing complex clinical data outside of a hospital or clinic environment, where queries cannot be addressed promptly. This problem is arguably more acute in glaucoma due to the large amount of technical detail provided in each VF report, interpretation of which requires specialist knowledge. One potential solution is to offer a high-level overview of results accessible via an online portal or downloadable app, although this would also require technical expertise and should be explored in further research. In addition to providing transparency regarding test results, such a service may encourage behavioural reinforcement to consistently complete VF testing at home, while also providing confirmation of successful completion of the test.

Our findings highlighted that greater responsibility through home testing was empowering for some participants, and may encourage a sustainable approach to living with glaucoma. However, two participants preferred conventional in-clinic VF testing rather than at home. In that respect, an assessment of individual self-efficacy regarding personal health management may help to prioritise individuals where home monitoring is most likely to be tolerated and successful. There may also be opportunity to encourage self-management behaviours through 'Expert Patient Initiatives' which have been shown to be successful in helping patients develop new health-related skills and promote active patient involvement in care.[44]

The present study explored experiences of home VF testing over 6 months; it remains unknown if favourable opinions and compliance persist long term. Yet, home monitoring would not necessarily be for life and might be most beneficial for the first 2 years following diagnosis.[33] The study was designed to collect monthly VF measurements, which is more frequent than typical VF monitoring, further research assessing patient interaction with glaucoma home monitoring over an extended time period is warranted. Our study measured VFs using a variant of the Eyecatcher test,[12 16] however there is no reason to expect that the findings would differ between other VF home monitoring systems.

The primary limitation of the present study is the fact that participants were self-selecting and, as a result, highly motivated and relatively homogeneous. All participants were members of a glaucoma charity organisation and were highly engaged with their eye health. The ability, and motivation, to undertake home perimetry, as well as the opinion of telemedicine in general, may be considerably lower in the general population. Further, while our sample included people with a range of ages, geographical locations and VF loss severities, it was comprised solely of Caucasian individuals who had the time and motivation to engage with research. We did not examine the impact of individual differences in socioeconomic status, lifestyle, health literacy, or various cultural or linguistic factors, each of which could impact the feasibility and acceptability of home VF testing substantively. Finally, due to the longitudinal nature of the research, our results may also be influenced by recall bias (ie, poor accuracy of participants' recollection); however, given the consistency in responses across interviews, we believe that the overall message of the report remains accurate.

In summary, our self-selecting participants with glaucoma were widely accepting and highly capable of home VF testing, and this may partly be explained by a combination of perceived benefits and motivations. Participants believed home monitoring may help to improve service

delivery, and Eyecatcher was compared positively with conventional perimetry. There are certain barriers to home VF monitoring such as unfamiliarity with bespoke testing devices and measurement features, yet these issues do not appear to be insurmountable. This report describes the experience of home monitoring among the target population. Details regarding patient preferences and attitudes can be used to inform future study design and guide current strategies to maximise the effectiveness of home monitoring programmes and help build a more sustainable glaucoma service infrastructure.

**Acknowledgements** The authors would like to thank all participants for their valuable contributions to the study. They would like to thank the following participants for their assistance during the member-check exercise, and who consented to be named: Richard Clarke; Jane de Swiet; Joan Dickson; Carolyn Game; Miles Holroyd; Christine Janner-Burgess; Stephanie Jones; Mick Kelsey; Graham Maynard; Sheila Page; Andrew Snell; Derek Warner; Mark Wagstaff.

**Contributors** LJ performed study design, data collection, data analysis and manuscript preparation. TC performed study design, data analysis and manuscript preparation. PC performed study design, data collection and manuscript critique. PRJ performed study design, hardware preparation and manuscript critique. DJT performed data analysis and manuscript critique. DSA performed hardware preparation and manuscript critique. DFE performed study design and manuscript critique. DPC performed study design and manuscript critique.

**Funding** This study was funded by a Fight for Sight (UK) project grant (#1854/1855) and by the International Glaucoma Association (IGA)/College of Optometrists 2019 Award (which is funded by the IGA and administered by the IGA in conjunction with the College of Optometrists). Author DSA was supported by the European Union's Horizon 2020 research and innovation programme under Marie Sklodowska-Curie grant agreement no. 675 033.

**Disclaimer** The funding organisations had no role in the design or conduct of this research. Any inaccuracies in the views presented are the responsibility of the authors.

**Competing interests** No conflicting relationship exists for any author. DPC reports unrestricted grants from Roche, Santen, Allergan; speaker fees from THEA, Bayer, Santen, Allergan; consultancy with Centervue; all outside the submitted work.

**Patient and public involvement** Patients and/or the public were involved in the design, or conduct, or reporting, or dissemination plans of this research. Refer to the Methods section for further details.

**Patient consent for publication** Not required.

**Ethics approval** The study was approved by the Ethics Committee for the School of Health Sciences, City, University of London (#ETH1819-0532), and carried out in accordance with the tenets of the Declaration of Helsinki.

**Provenance and peer review** Not commissioned; externally peer reviewed.

**Data availability statement** No data are available.

**ORCID iDs**
Lee Jones http://orcid.org/0000-0002-8030-1211
Deanna J Taylor http://orcid.org/0000-0001-8261-5225

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
