## [Reviewer comments · BMJ Open]

ARTICLE DETAILS

TITLE (PROVISIONAL)	Acceptability of a home-based visual field test (Eyecatcher) for glaucoma home monitoring: A qualitative study of patients' views and experiences.
AUTHORS	Jones, Lee; Callaghan, Tamsin; Campbell, Peter; Jones, Pete; Taylor, Deanna; Asfaw, Daniel; Edgar, David; Crabb, David

VERSION 1 – REVIEW

REVIEWER	Dr. Mohan Krishna Shrestha Tilganga Institute of Ophthalmology, Nepal
REVIEW RETURNED	13-Oct-2020

GENERAL COMMENTS	The reviewer completed the checklist but made no further comments.
--

REVIEWER	Thomas Chengxuan Lu School of Medicine, University of New South Wales. Australia
REVIEW RETURNED	03-Nov-2020

GENERAL COMMENTS	This is a qualitative study undertaken to determine participant experiences of home visual field testing (Eyecatcher) featuring 20 glaucoma patients. Visual fields were measured at home once/month for a period of 6 months. Eye catcher is a very innovative, open source software for glaucoma detection, utilising tablet computer which was delivered to each participant. Median age was 71, which is well-representative of glaucoma patients. Great that feedback was sought from participants to inform the thematic analysis and such high percentage responded. Overall very interesting study. Major Comments: 1. The sample population here seems to be a homogenous, highly motivated population group with only Caucasian participants. Although I do understand that this sample responded to an advertisement. Markers of socioeconomic status, health literacy and language are not recorded. It is highly likely that this sample is not representative of the normal population group of glaucoma patients which limits the generalisability of the data. The ability to operate and the motivation to undertake home perimetry is much likely lower in the general population, particularly from participants from linguistically diverse backgrounds. This should be addressed within the manuscript.2. Some themes overlap, and there is a diversity of opinions among participants regarding certain aspect of the Eyecatcher program. For example, the removal of perimetry bowl formulated a sense of freedom, but lead to an increased sense of distraction from
---

	the environment. Yet conducting the test away from the hospital lead to less distraction. Similarly, the response is also mixed for the visual feedback function, length of the Eyecatcher tests etc. This should be acknowledged more clearly within the manuscript. 3. There is a lot of emphasis on indirect costs associated with hospital visit which is true. However, there is no information provided on the costs associated with home perimetry testing. 4. How is data going to be useful for the development of home perimetry monitoring? 5. Are there any conflicts of interest in this work? Although the algorithm is Open Source, are the authors involved in the creation and formulation of Eyecatcher as a practical program? Are there any benefits and interests to be attained by the authors? This should be declared. Minor comments: Page 7: Line 5-6 – why was only ten participants randomly selected to practise Eyecatcher? Explanation should be provided. This would have affected thematic outcomes due to a heterogeneous user experience. Page 5: Line 32-33 – what does long saturated HES mean? Is there a reference for this? Page 14: Line 59 – can the authors clarify the definition of ‘older people’ Page 27: Line 21 – what insurmountable barriers are these?
--	---

REVIEWER	Amanda Bicket University of Michigan Kellogg Eye Center Ann Arbor, MI United States of America
REVIEW RETURNED	05-Jan-2021

GENERAL COMMENTS	The objective of this author team is to report qualitative data describing the acceptability of home visual field monitoring in people with glaucoma. Overall, I think this manuscript is well-done and presents important information that nicely complements the investigators’ AJO publication describing the accuracy of the Eyecatcher. I feel it will be suitable for publication once the issues listed below are addressed. My primary criticism is that its Discussion presents the views of a limited, highly self-selected population (those who read the Glaucoma UK newsletter and volunteered for this study) as representative of the wider glaucoma population; I’ve elaborated below. Abstract: - The abstract is clear and complete, but might benefit from some detail about the interviews (whether they were conducted in person or via video/telephone conference) and interviewees (type(s) and severity of glaucoma). - While the “Setting” section acknowledges that all participants volunteered in response to an advertisement, the “Conclusions” should also acknowledge that this was a highly self-selecting group of patients who not only read the Glaucoma UK newsletter but wanted to be interviewed about Eyecatcher. The statement “glaucoma patients were widely accepting and highly capable of home VF testing” seems over-reaching; adding “the population of glaucoma patients who volunteered to be interviewed” or something
---

	similar would help Introduction: - The Introduction nicely motivates the report and explains its context – the saturated HES system – which is relatable for providers in many other parts of the world as well. It is reassuring that the investigators have also “assessed the accuracy of home VF monitoring in a parallel investigation;” as the full citation for that report in AJO is now available, the References should be edited to include it. Methods: - The investigators did an excellent job of describing their data collection and analysis methods, making them accessible to an audience (eye and vision researchers and clinicians) which may not be uniformly familiar with qualitative research. I particularly appreciated their description of their member-checking exercise and “patient and public involvement.” I have no edits for this section. Results: - As is typically the case in reports of qualitative research, this section is lengthy. I would suggest shortening where possible by including only 1-2 quotes per observation. Some suggestions follow; authors should not feel obliged to take all of them: o Theme 1: p. 12 lines 9-26 might require only 2 quotes, rather than 3; p. 10 lines 43-56 and p.11 lines 11-26 might be captured with single quotes o Theme 3: p. 16 lines 44-54 and p. 18 lines 19-29 might be captured with single quotes o Theme 4: p. 19-20 might require 2 quotes, rather than 4; p. 18 lines 43-55, p. 19 lines 13-29, and p. 20 lines 39-52 might be captured with single quotes Discussion: - The second sentence, “Our findings suggest that use of a home-based perimeter for VF testing was feasible and widely accepted among people with glaucoma,” is misleading and not entirely consistent with the goals of qualitative research, which is not meant to bin or simplify perspectives, but to capture the complexity of individual experiences, preferences, etc. It is meaningful that the glaucoma patients who sought out participation in this study by responding to an advertisement in the UK Glaucoma newsletter feel positive and confident about home VF monitoring, because providers should know that such patients are out there, excited to have this tool at their disposal. However it is not reasonable to expect all glaucoma patients to embrace this technology. It is mentioned among the limitations, but I feel the framing of the Discussion (along with the Conclusions in the abstract) should be modified to acknowledge this important limitation. - The Discussion is quite long and I feel should be streamlined. For instance, paragraphs 6-8 (on cost) could be condensed. - If other themes – like testing conditions, which were mentioned by 90% or participants (paragraph 9) - appeared in the vast majority (or, conversely, the small minority) of interviews, that information would help readers understand how the study population as a whole discussed the themes described. Those data (proportions or percentages) should be included in both Results and Discussion.
--	---

VERSION 1 – AUTHOR RESPONSE

Reviewer:1

Dr. Mohan Shrestha, Tilganga Institute of Ophthalmology, Public Health Development , Nepal
Comments to the Author:
None

Reviewer:2

Dr. Thomas Lu, University of New South Wales Comments to the Author:
I think this is a good study that is worth-while publishing in parallel to the quantitative that is undertaken for Eyecatcher. I have attached some comments. The major issue is the generalisability of this data, as it would seem that many of the participants are all highly motivated. This is a qualitative study undertaken to determine participant experiences of home visual field testing (Eyecatcher) featuring 20 glaucoma patients. Visual fields were measured at home once/month for a period of 6 months. Eye catcher is a very innovative, open source software for glaucoma detection, utilising tablet computer which was delivered to each participant. Median age was 71, which is well-representative of glaucoma patients. Great that feedback was sought from participants to inform the thematic analysis and such high percentage responded. Overall very interesting study.

We thank the reviewer for their insightful and positive comments. In the revised manuscript we have addressed the major and minor comments raised and we have responded to each comment as follows:

Major Comments:

1. The sample population here seems to be a homogenous, highly motivated population group with only Caucasian participants. Although I do understand that this sample responded to an advertisement. Markers of socioeconomic status, health literacy and language are not recorded. It is highly likely that this sample is not representative of the normal population group of glaucoma patients which limits the generalisability of the data. The ability to operate and the motivation to undertake home perimetry is much likely lower in the general population, particularly from participants from linguistically diverse backgrounds. This should be addressed within the manuscript.

We agree that our findings do not suggest VF home monitoring will be tolerated and successful in all populations. We have now amended the manuscript to reflect that our participants were a highly selective sample from the glaucoma community.

Abstract (Conclusions): *'Participants identified a broad range of benefits to VF home monitoring and discussed areas for service improvement. Eyecatcher was compared positively to conventional VF testing using HFA. Home monitoring may be acceptable to at least a subset of people with glaucoma.'*

Discussion (Paragraph 1): *'Self-selecting members of a glaucoma patient organisation recognised the usefulness of testing and valued the opportunity to have their condition monitored from home. The findings suggest some individuals with glaucoma may embrace wider implementation of VF home monitoring.'*

Discussion (Paragraph 14): *'The primary limitation of the present study is the fact that participants were self-selecting and, as a result, highly motivated and relatively homogenous. ~~sample consisted of self-selecting volunteers who may be different to, for example, less motivated individuals which may limit generalisability.~~ All participants were members of a glaucoma charity organisation and were ~~therefore a highly selective group who may differ from those less~~ highly engaged with their eye health. The ability, and motivation, to undertake home perimetry, as well as the opinion of telemedicine in general, may be considerably lower in the general population. Further, while our sample included people with a range of ages,*

geographic locations, and VF loss severities, it was comprised solely of Caucasian individuals who had the time and motivation to engage with research. We did not examine the impact of individual differences in socioeconomic status, lifestyle, health literacy, or various cultural or linguistic factors, each of which could impact the feasibility and acceptability of home VF testing substantively.'

Discussion (Paragraph 15): *'In summary, ~~people with glaucoma~~ our self-selecting glaucoma participants were widely accepting and highly capable of home VF testing, and this may partly be explained by a combination of perceived benefits and motivations.'*

2. Some themes overlap, and there is a diversity of opinions among participants regarding certain aspect of the Eyecatcher program. For example, the removal of perimetry bowl formulated a sense of freedom, but lead to an increased sense of distraction from the environment. Yet conducting the test away from the hospital lead to less distraction. Similarly, the response is also mixed for the visual feedback function, length of the Eyecatcher tests etc. This should be acknowledged more clearly within the manuscript.

We have amended the manuscript as follows:

Discussion (Paragraph 4): *'Participants generally compared Eyecatcher positively to conventional perimetry. In particular, the absence of chin and head restraints with Eyecatcher was particularly preferred over the HFA, which was viewed as an uncomfortable and unnatural testing procedure. There was a diversity of opinions, however, and some participants did question whether Eyecatcher's more relaxed testing procedure, and/or the lack of a conventional perimetric bowl (to occlude potential distractors), may result in more distractions and less reliable measurements. Our quantitative analysis suggests the home monitoring data were generally of good quality, and that any loss of reliability was largely compensated for by the overall increase in the volume of data.⁽¹⁵⁾ However, distractions will always be a concern for any unsupervised/uncontrolled testing. In response to this we have developed computer-vision, AI technology to autonomously detect and account for lapses in concentration. This technology uses data from the web cam on the portable device and can be embedded within Eyecatcher. Our investigation of this technology in a large group of people, albeit with healthy vision, is the subject of a report published elsewhere.⁽³⁰⁾ More generally, the fact that there were mixed opinions regarding the Eyecatcher interface (i.e., with some finding the removal of the bowl liberating, while others expressed concerns regarding increased distractions), highlights the inherent complexities associated with developing and implementing novel health-related technologies and ensuring they match the needs and preferences of the target users. We hope that the in-depth feedback generated in this study will serve as a useful reference for clinicians, researchers and commissioners working in the area of VF home monitoring.'*

3. There is a lot of emphasis on indirect costs associated with hospital visit which is true. However, there is no information provided on the costs associated with home perimetry testing.

In line with comments from Reviewer 3, we have significantly augmented the discussion to include fewer details about direct/indirect costs of hospital visits/perimetry. We have expanded on costs associated with home VF testing:

Discussion (Paragraph 6): *'Conversely, it should be noted that home-monitoring itself has associated costs, and the extent to which home monitoring of VFs is cost-effective is yet to be established.⁽³²⁾ Prospective costs associated with VF home monitoring include data management and the cost of supplying and maintaining any associated hardware (though exactly what hardware is required, or whether a perimetry test could even be provided simply as a 'downloadable app' remain unresolved questions, and highly contentious). Home monitoring provides opportunity for increased frequency of VF testing, which may place a burden on clinicians. Considering previous estimations of costs associated with conventional VF testing,⁽³³⁾ we predict home VF testing will prove cost-effective, at least for the most high-*

risk patients (e.g., younger individuals, or those with risk factors for fast progression). A formal economic evaluation is required, however.'

4. How is data going to be useful for the development of home perimetry monitoring?

We have now added details on how our findings will be helpful for developing wider VF home monitoring programmes:

Discussion (Paragraph 4): 'We hope that the in-depth feedback generated in this study will serve as a useful reference for clinicians, researchers and commissioners working in the area of VF home monitoring.'

Discussion (Paragraph 15): 'This report describes the experience of home monitoring among the target population. Details regarding patient preferences and attitudes can be used to inform future study design and guide current strategies to maximise the effectiveness of home monitoring programmes and help build a more sustainable glaucoma service infrastructure.'

5. Are there any conflicts of interest in this work? Although the algorithm is Open Source, are the authors involved in the creation and formulation of Eyecatcher as a practical program? Are there any benefits and interests to be attained by the authors? This should be declared.

We are currently exploring all possibilities (commercial and non-commercial) for developing Eyecatcher into a distributable "product". However, at present we have no specific plans. We have made the following addition to the manuscript:

'To date, the development of Eyecatcher has been funded in part by university research time and non-commercial research grants. The complete source code is available online (<https://github.com/petejonze/eyecatcher>) under a non-commercial (GPL 3.0) license.'

Minor comments:

Page 7: Line 5-6 – why was only ten participants randomly selected to practise Eyecatcher? Explanation should be provided. This would have affected thematic outcomes due to a heterogeneous user experience.

We agree that exposing a proportion of the participants to a practice with Eyecatcher has potential to impact user experience. However, we do not believe this will have impacted the overall outcomes of the study. For example, during the trial participants from both groups contacted the study teams with questions about the device and the Eyecatcher testing programme. In addition, both groups of participants described similar experiences during the interviews on their exit from the trial. The primary reason for providing half of participants with an initial practice was to determine perimetric learning effects using Eyecatcher. We now make this clear in the manuscript:

Methods (Paragraph 2): 'Ten participants (50%) were randomly selected to practice Eyecatcher once in each eye, under supervision at baseline (Table 1.). The rationale for dividing the sample in this way was to determine whether there were perimetric learning effects using Eyecatcher. We report in our parallel investigation there was no significant difference in mean absolute error between the eyes of participants who received a practice, and those who did not.'⁽¹⁵⁾

Page 5: Line 32-33 – what does long saturated HES mean? Is there a reference for this?

We have amended this sentence and included references:

Introduction (Paragraph 3): 'Home monitoring, whereby disease progression is proactively monitored through patient-led, home-based testing, may help alleviate the increasingly unsustainable burden on hospital eye services ⁽⁸⁻⁹⁾ ~~may offer a solution to the long-saturated HES.~~

Page 14: Line 59 – can the authors clarify the definition of 'older people'

We have now amended this sentence to refer to those less familiar with technology, rather than older people. We have also provided examples of 'older age' in the discussion:

Results: *'In some cases insufficient training in using Eyecatcher resulted in a lack of confidence and apprehension around the test, and this was considered a barrier to home monitoring, particularly for those who may be less familiar with technology'*

Discussion (Paragraph 11): *'Patterns of technology usage suggest older ophthalmic patients frequently use technology, for example, 88.5% of people aged 66-79 years attending Manchester Royal Eye Hospital report regularly using technologies including desktop and laptop computers ⁽³⁹⁾. Many older adults aged 65 and over use devices relating to healthcare, such as glucose and cardiac monitoring tools.⁽⁴⁰⁾ It is perhaps most appropriate to expect large variability in abilities relating to technology amongst older populations.'*

Page 27: Line 21 – what insurmountable barriers are these?

We have now amended this sentence, as follows:

Discussion (Paragraph 15): *'There are certain barriers to home VF monitoring such as unfamiliarity with bespoke testing devices and measurement features, yet these issues do not appear to be insurmountable.'*

Reviewer:3

Dr. Amanda Bicket, University of Michigan Comments to the Author: The objective of this author team is to report qualitative data describing the acceptability of home visual field monitoring in people with glaucoma. Overall, I think this manuscript is well-done and presents important information that nicely complements the investigators' AJO publication describing the accuracy of the Eyecatcher. I feel it will be suitable for publication once the issues listed below are addressed. My primary criticism is that its Discussion presents the views of a limited, highly self-selected population (those who read the Glaucoma UK newsletter and volunteered for this study) as representative of the wider glaucoma population; I've elaborated below.

We thank the reviewer for their thoughtful and supportive comments. In the revised manuscript we have addressed the comments raised and we have responded to each comment as follows:

Abstract:

The abstract is clear and complete, but might benefit from some detail about the interviews (whether they were conducted in person or via video/telephone conference) and interviewees (type(s) and severity of glaucoma).

We have now added these details to the abstract.

Design: *Qualitative study using face-to-face semi-structured interviews. Transcripts were analysed using thematic analysis.*

Participants: *Twenty adults (10 female; median age 71 years) with a diagnosis of glaucoma were recruited (including open angle and normal tension glaucoma; Mean deviation = 2.5 to -29.9 dB).*

While the “Setting” section acknowledges that all participants volunteered in response to an advertisement, the “Conclusions” should also acknowledge that this was a highly self-selecting group of patients who not only read the Glaucoma UK newsletter but wanted to be interviewed about Eyecatcher. The statement “glaucoma patients were widely accepting and highly capable of home VF testing” seems over-reaching; adding “the population of glaucoma patients who volunteered to be interviewed” or something similar would help.

We agree it is important to make this distinction in the abstract. We have amended as follows:

Conclusions: *Participants identified a broad range of benefits to VF home monitoring and discussed areas for service improvement. Eyecatcher was compared positively to conventional VF testing using HFA. Home monitoring may be acceptable to at least a subset of people with glaucoma.*

Introduction:

The Introduction nicely motivates the report and explains its context – the saturated HES system – which is relatable for providers in many other parts of the world as well. It is reassuring that the investigators have also “assessed the accuracy of home VF monitoring in a parallel investigation;” as the full citation for that report in AJO is now available, the References should be edited to include it.

We have now included the updated AJO citation, with thanks.

Methods:

The investigators did an excellent job of describing their data collection and analysis methods, making them accessible to an audience (eye and vision researchers and clinicians) which may not be uniformly familiar with qualitative research. I particularly appreciated their description of their member-checking exercise and “patient and public involvement.” I have no edits for this section.

We thank the reviewer for this feedback.

Results:

- As is typically the case in reports of qualitative research, this section is lengthy. I would suggest shortening where possible by including only 1-2 quotes per observation. Some suggestions follow; authors should not feel obliged to take all of them:

- o Theme 1: p. 12 lines 9-26 might require only 2 quotes, rather than 3; p. 10 lines 43-56 and p.11 lines 11-26 might be captured with single quotes
- o Theme 3: p. 16 lines 44-54 and p. 18 lines 19-29 might be captured with single quotes
- o Theme 4: p. 19-20 might require 2 quotes, rather than 4; p. 18 lines 43-55, p. 19 lines 13-29, and p. 20 lines 39-52 might be captured with single quotes

Thank you. As suggested, we have removed quotations in sections recommended by the reviewer (eight quotes removed in total).

Discussion:

-The second sentence, "Our findings suggest that use of a home-based perimeter for VF testing was feasible and widely accepted among people with glaucoma," is misleading and not entirely consistent with the goals of qualitative research, which is not meant to bin or simplify perspectives, but to capture the complexity of individual experiences, preferences, etc. It is meaningful that the glaucoma patients who sought out participation in this study by responding to an advertisement in the UK Glaucoma newsletter feel positive and confident about home VF monitoring, because providers should know that such patients are out there, excited to have this tool at their disposal. However it is not reasonable to expect all glaucoma patients to embrace this technology. It is mentioned among the limitations, but I feel the framing of the Discussion (along with the Conclusions in the abstract) should be modified to acknowledge this important limitation.

Similar comments were raised by reviewer 2, we have now tried to balance the manuscript to ensure the message is clear that visual field home monitoring may be successful among a highly selective glaucoma population.

Abstract (Conclusions): *'Participants identified a broad range of benefits to VF home monitoring and discussed areas for service improvement. Eyecatcher was compared positively to conventional VF testing using HFA. Home monitoring may be acceptable to at least a subset of people with glaucoma.'*

Discussion (Paragraph 1): *'Self-selecting members of a glaucoma patient organisation recognised the usefulness of testing and valued the opportunity to have their condition monitored from home. The findings suggest some individuals with glaucoma may embrace wider implementation of VF home monitoring.'*

Discussion (Paragraph 14): *'The primary limitation of the present study is the fact that participants were self-selecting and, as a result, highly motivated and relatively homogenous. ~~sample consisted of self-selecting volunteers who may be different to, for example, less motivated individuals which may limit generalisability.~~ All participants were members of a glaucoma charity organisation and were ~~therefore a highly selective group who may differ from those less highly engaged with their eye health.~~ The ability, and motivation, to undertake home perimeter, as well as the opinion of telemedicine in general, may be considerably lower in the general population. Further, while our sample included people with a range of ages, geographic locations, and VF loss severities, it was comprised solely of Caucasian individuals who had the time and motivation to engage with research. We did not examine the impact of individual differences in socioeconomic status, lifestyle, health literacy, or various cultural or linguistic factors, each of which could impact the feasibility and acceptability of home VF testing substantively. ~~Our sample was also specific to those with capacity to read and respond to the recruitment advertisement and testing instructions, as well as, for example, storing the hardware for the duration of the study. Notwithstanding this, our sample consisted of people of different disease severity and from different geographic locations.'~~*

Discussion (Paragraph 15): *'In summary, ~~people with glaucoma~~ our self-selecting glaucoma participants were widely accepting and highly capable of home VF testing, and this may partly be explained by a combination of perceived benefits and motivations.'*

The Discussion is quite long and I feel should be streamlined. For instance, paragraphs 6-8 (on cost) could be condensed.

We have now reduced the discussion by removing most of paragraph 7 and significantly condensing paragraph 6 and 8.

If other themes – like testing conditions, which were mentioned by 90% of participants (paragraph 9) - appeared in the vast majority (or, conversely, the small minority) of interviews, that information would help readers understand how the study population as a whole discussed the themes described. Those data (proportions or percentages) should be included in both Results and Discussion.

We chose to report that 90% of participants mentioned testing conditions to highlight that participants were anxious about doing the test right/wrong; however it was gratifying that these aspects of VF home monitoring (e.g. ambient illumination / exact viewing distance) do not appear to be as important as expected (as identified in our parallel quantitative manuscript). This outcome highlights it will be crucial to give clear guidance to users, or modify the equipment to obviate the problem (e.g. automatic detection of ambient lighting / viewing distance).

We feel that reporting proportions/percentages for the identified themes/subthemes may unwillingly encourage readers to rank the findings with regard to their importance. We agree with the reviewer's earlier comment that the goal of qualitative research should not be to simplify/quantify perspectives, but rather capture and display the broader details and experiences of all participants.